# Euthanasia of Cattle: Practical Considerations and Application

**DOI:** 10.3390/ani8040057

**Published:** 2018-04-17

**Authors:** Jan Keith Shearer

**Affiliations:** Professor and Extension Veterinarian, College of Veterinary Medicine, Iowa State University, Ames, IA 50011-1250, USA; JKS@iastate.edu

**Keywords:** euthanasia, carcass disposal, firearms for euthanasia, captive bolt, anatomic sites for euthanasia, mass depopulation, euthanasia, caring and killing paradox, compassion fatigue, post-traumatic stress disorder, perpetuation-induced traumatic stress

## Abstract

**Simple Summary:**

Methods recognized as acceptable for the euthanasia of cattle include overdose of an anesthetic, gunshot and captive bolt. The most common injectable anesthetic agent used for euthanasia is pentobarbital and while it may be the preferred method for euthanasia in sensitive situations, it creates significant challenges for disposal of animal remains. Gunshot and captive bolt are the more common methods used on farms and ranches because they are inexpensive, humane and do not complicate carcass disposal. Firearms must be of the proper caliber and loaded with the proper ammunition. Captive bolt, equipped with a penetrating bolt, is to be used on adult animals, whereas the non-penetrating (mushroom head) bolt should be reserved for use in calves (three months of age or less). In addition to selection of the proper firearm or captive bolt, successful euthanasia requires use of the proper anatomic site and adjunctive steps to assure death. The indicators of unconsciousness and death must be clearly understood and confirmed in all situations involving euthanasia. Tools for the efficient depopulation of a large feedlot, dairy or beef cattle operation as may be required in a national animal health emergency situation have been developed and validated as effective. Finally, the human impact of euthanasia cannot be underestimated. Symptoms of mental illness including depression, grief, sleeplessness and destructive behaviors including alcoholism and drug abuse are not uncommon for those who participate in the euthanasia of animals.

**Abstract:**

Acceptable methods for the euthanasia of cattle include overdose of an anesthetic, gunshot and captive bolt. The use of anesthetics for euthanasia is costly and complicates carcass disposal. These issues can be avoided by use of a physical method such as gunshot or captive bolt; however, each requires that certain conditions be met to assure an immediate loss of consciousness and death. For example, the caliber of firearm and type of bullet are important considerations when gunshot is used. When captive bolt is used, a penetrating captive bolt loaded with the appropriate powder charge and accompanied by a follow up (adjunctive) step to assure death are required. The success of physical methods also requires careful selection of the anatomic site for entry of a “free bullet” or “bolt” in the case of penetrating captive bolt. Disease eradication plans for animal health emergencies necessitate methods of euthanasia that will facilitate rapid and efficient depopulation of animals while preserving their welfare to the greatest extent possible. A portable pneumatic captive bolt device has been developed and validated as effective for use in mass depopulation scenarios. Finally, while most tend to focus on the technical aspects of euthanasia, it is extremely important that no one forget the human cost for those who may be required to perform the task of euthanasia on a regular basis. Symptoms including depression, grief, sleeplessness and destructive behaviors including alcoholism and drug abuse are not uncommon for those who participate in the euthanasia of animals.

## 1. Introduction

Euthanasia is the term used to describe ending the life of an animal in a way that minimizes or eliminates pain and distress. The way an animal dies is an important aspect of its welfare. Assuring a humane end is the ethically responsible thing to do regardless of whether the animal is a rat, a dog or a cow. Rollin writes that “*involuntary euthanasia is not only permissible for suffering animals, but indeed obligatory*” [1]. Uncontrollable pain and suffering offers justification for euthanasia that is easier for most to accept. The rationalization for killing healthy animals outside of the context of needs for food, medical research, disease control and management of pet overpopulation (e.g., for recreational purposes such as hunting purely for sport) is more difficult.

In food producing animals, the loss of productive function resulting from disease or injury presents at least two options: slaughter or euthanasia. Slaughter should only be considered for animals that are not in pain, freely able to stand and walk, capable of being transported and without disease, drug, or chemical residue that might constitute a public health risk. Euthanasia is the appropriate choice whenever the above conditions cannot be met. The life of an animal may be spared from slaughter or euthanasia if there is reasonable probability that treatment would promote improvement to a healthy state.

This article is intended to provide a general review of euthanasia procedures in cattle. Specific issues discussed include acceptable versus unacceptable methods of euthanasia; advantages and disadvantages to the use barbiturates; use of the physical methods of gunshot and captive bolt; carcass disposal considerations; anatomic site selection for use of a firearm or captive bolt; visible indicators of unconsciousness and secondary (i.e., adjunctive) methods to assure death for the euthanasia of cattle. The possibility of a national health emergency such as an outbreak of Foot and Mouth Disease (FMD) may necessitate a mass depopulation strategy capable of the euthanasia of large numbers of animals. In preparation for the possibility of such an event, United States Department of Agriculture, Animal and Plant Health Inspection Service (USDA-APHIS) has developed and validated a portable pneumatic penetrating captive bolt that does not require an adjunctive step to assure death. Two types of bolts were studied; one with low-pressure air channel pithing through the bolt and another without the air channel-pithing feature. Results of these studies are briefly reviewed.

A discussion of euthanasia would not be complete without consideration of its effects on the emotional well-being of those required to routinely perform these procedures. Much has been written about veterinarians and technicians in veterinary practice, animal shelters and research laboratories, but the struggles of those engaged in food animal work are poorly documented in the literature. Although raised with the knowledge they will one day be slaughtered for human use does not mean that it is easier to euthanize them if the need should arise. Anecdotal and research information indicates that euthanasia can have significant effects on human mental health.

Whether one be a veterinarian, livestock owner or farm employee, attention to the details of properly executing euthanasia procedures is important for assuring a humane end to life in animals and reduces the potential for those even worse experiences when a failed attempt ends in extreme animal pain and distress.

## 2. Acceptable Verses Unacceptable Methods of Euthanasia

Not all methods of ending life fit the definition of euthanasia. In order to provide direction to veterinarians and others who may be required to euthanize animals, the AVMA’s Panel on Euthanasia classified methods as “acceptable”, “acceptable with conditions”, and “unacceptable” [2]. Acceptable methods are described as those that consistently produce a humane death when used as the sole means of euthanasia. Pentobarbital overdose fits this description of an acceptable method of euthanasia. Methods described as “acceptable with conditions” are those that may require certain conditions be met to consistently produce a humane death; for example, selection of the proper anatomic site, proper caliber of firearm and type of bullet or powder charge for captive bolt. Methods deemed “unacceptable” are those that are considered to be inhumane under any conditions [2].

## 3. Unacceptable Methods of Euthanasia

Unacceptable methods of euthanasia are those that fail to meet the requirements of euthanasia. In cattle these would include: (1) manually applied blunt force trauma (as with a large hammer), (2) injection of chemical agents or other substances not specifically designed or labeled for euthanasia (i.e., disinfectants, cleaning solutions, etc.), (3) air injection into the vein, (4) electrocution as with a 120 volt electrical cord, (5) drowning, (6) exsanguination of conscious animals, and (7) deep tranquilization as with xylazine or other alpha-2 agonist followed by potassium chloride or magnesium sulfate [2]. Several of these methods are known to result in an inhumane death and for others the level of pain or distress associated with these methods is unknown.

Euthanasia of an animal by use of firearm or captive bolt may be bothersome, particularly for those less familiar with the physical methods of gunshot and captive bolt. Some, if not many, would much prefer to use an injectable euthanasia agent or other less violent method such as an elevated dose of xylazine hydrochloride (XH) followed by the rapid administration of KCl to cause cardiac arrest and death. A study conducted at Iowa State University in six yearling crossbred feedlot calves ranging in weight from 227–363 kg (500–800 lb) found that despite being administered 1000 mg of XH (i.e., nearly 10–20 times the normal dose); none of the animals achieved an anesthetic state based upon electroencephalogram (EEG) [3]. The position of the AVMA is as that stated in Goodman and Gilman’s Pharmacological Basis of Therapeutics, 11th Edition [4]: “Although large doses of alpha-2 agonists can produce a state resembling general anesthesia, they are recognized as being unreliable for that purpose.” Since an animal administered XH does not reach an unconscious (anesthetized) state, the use of KCl (which causes cardiac arrest) secondary to tranquilization with XH, does not meet the requirements of euthanasia and results in an inhumane death. The combination of XH followed by intravenous administration of KCl is an unacceptable method of euthanasia [2,4].

## 4. Acceptable Methods of Euthanasia

The acceptable methods of euthanasia in cattle include anesthetic overdose and the physical methods of gunshot and penetrating and non-penetrating captive bolt. The modes of action are different for each [2].

### 4.1. Anesthetic Overdose

Barbiturates and barbituric acid derivatives. Administered by intravenous injection, barbiturates induce a smooth transition from consciousness to unconsciousness and death by causing depression of the central nervous system and respiratory centers in the brain leading to cardiac arrest. Despite their effectiveness as euthanasia agents, there are several drawbacks to their use including cost, the need for intravenous administration of the drug, the necessity to maintain a careful accounting of amounts used, regulatory requirements specifying that only a veterinarian administer these agents and residues that limit carcass disposal options [2]. Nonetheless, depending upon circumstances, the use of a barbiturate overdose may be most the appropriate, particularly in sensitive situations.

### 4.2. The Physical Methods: Firearms and Captive Bolt

Traumatic brain injury using physical methods. The types of traumatic brain injury (TBI) normally encountered are broadly characterized as blunt, non-missile and missile [5,6]. Non-missile injury is that commonly encountered by humans in auto accidents or animals hit by cars. Injury to the brain results from the rapid acceleration and deceleration of the brain along with rotational and sheering forces within the cranial cavity. The blunt form of TBI is also a non-missile type injury and similar to that inflicted by non-penetrating captive bolt [7]. Proper placement resulted in an immediate loss of consciousness, a localized fracture of the frontal bone of the skull, widespread hemorrhage beneath the impact site and in the temporal and frontal lobes of the brain and brainstem. Researchers concluded that the concussive forces were sufficient to cause unconsciousness and a rapid ventro-caudal displacement of the brain on impact [7].

Traumatic brain injury occurring from a missile type of projectile (i.e., bullet or bolt from a captive bolt) are characterized as penetrating or perforating lesions [5,6]. The extent of TBI is determined by speed of the bullet and the energy it displaces to adjacent tissues. In the case of penetrating wounds, the projectile (bullet or bolt) enters the cranial cavity but does not exit. Perforating wounds are those whereby the projectile not only penetrates through the skull, but if properly directed will traverse the brain and exit at some other site. Some bullets (i.e., hollow points) are designed to fragment on impact. Fragmentation of the bullet will reduce penetration as the bullet’s energy is dissipated on impact. If the bullet penetrates the skull and then fragments multiple secondary wound tracks will be produced increasing brain damage. Similarly, brain destruction is increased when the bullet (or the bolt from a captive bolt) drives shards of bone deep into the brain along its pathway. High-velocity bullets tend to produce perforating injuries, whereas lower velocity bullets produce penetrating injury and retention of the bullets within the skull [6].

### 4.3. Recommendations on Firearms for Euthanasia

“Properly applied, euthanasia by either gunshot or penetrating captive bolt, causes less fear and anxiety and induces a more rapid, painless, and humane death than can be achieved by most other methods”.[8]

Handguns or pistols are short-barreled firearms that may be fired with one hand. For the purposes of euthanasia, accuracy is improved when handguns are limited to close-range shooting within one meter (3.3 ft) or less of the intended target. Calibers ranging from 0.32 to 0.45 are recommended for euthanasia of cattle [2,9]. Although hollow point bullets will cause more damage compared with solid-point bullets they may not traverse the skull. However, as described above, hollow point bullets are designed to expand and fragment on impact with their targets, which reduces depth of penetration.

The 0.22 caliber handgun is not recommended for routine euthanasia of cattle regardless of the type of bullet used, because of the inability to consistently achieve desirable muzzle energies with standard commercial loads [2]. Muzzle velocity and energy of a bullet are generally higher when fired from a rifle. A longer barrel allows the propellant (gunpowder) to burn more completely which maximizes velocity of the bullet as it leaves the muzzle. The shorter barrel of a handgun results in reduced muzzle velocity since much of the pressure propelling the bullet through the barrel is dissipated into the air prior to complete burning of the gunpowder.

Thomson et al. found that the 9 mm pistol firing a metal-jacketed bullet performed poorly in a study involving bovine cadaver skulls [10]. Despite an estimated kinetic energy (KE) of 427.85 Joules (J) with this firearm and bullet combination, it caused the least amount of brain tissue and brainstem trauma. One might speculate that a shorter barrel length may be part of the explanation. At least one source indicates that heavier rounds with larger propellant loads such as those used in 9 mm and 0.45 caliber pistols require a barrel length of 16 inches or more for bullets to reach maximum muzzle velocity [11]. Ultimately, the damage potential of a firearm and bullet/shotshell combination is determined by multiple factors including thickness of the skin and skull as well as characteristics of the projectile (i.e., hollow point or a full metal jacket) [10].

Rifles are long-barreled firearms that are usually fired from the shoulder. Unlike the barrel of a shotgun, which has a smooth bore for shot shells, the bore of a rifle barrel contains a series of helical grooves (called rifling) that cause the bullet to spin as it travels through the barrel. Rifling imparts stability to the bullet and improves accuracy. Rifles are the preferred firearm when it is necessary to shoot from a distance because they are capable of delivering bullets at much higher muzzle velocities and energies. General recommendations on rifle selection for use in euthanasia of cattle suggest use of a 0.22 magnum, or higher caliber. In the study by Thomson et al. [10] conducted with cadaveric feedlot steer skulls researchers found that of seven firearm ammunition combinations, the rifle-fired 0.22 Long Rifle (LR) with a hollow-point bullet and the pistol-fired 9 mm full metal jacketed round caused the least amount of brain and brainstem trauma. Researchers concluded that based upon the results of this study, these firearm ammunition combinations were poor choices for the euthanasia of feedlot cattle [10]. Similar results were obtained using 0.22 LR standard velocity 350 m/s (1148 ft/s) and 0.22 LR high velocity 382 m/s (1253 ft/s) bullets in a Canadian study of intact cattle skulls. Cadaver skulls were positioned 25 m (27.3 yd) from the shooter and oriented as anticipated would be the case in a mass depopulation scenario. Penetration of the bullet was unsatisfactory with both the 0.22 LR standard and high velocity bullets [12].

Shotguns loaded with buckshot or birdshot (#4 shot) are appropriate for euthanasia of cattle when used from a distance of three meters (3.3 yd) [2,10]. Although all shotguns are lethal at close range, the preferred gauges for euthanasia of mature cattle are 20, 16 or 12. The lead shot (i.e., small pellets) BBs in birdshot exit the barrel as compact bolus or mass with ballistic characteristics on impact with the skull at close range that are similar to those of a solid lead bullet. Penetration of the skull is assured with massive destruction of brain tissue from the dispersion of birdshot into the brain that results in immediate loss of consciousness and rapid death. A shotgun loaded with birdshot is less effective when shooting from a distance because the BBs in a shotshell will begin to scatter (i.e., spread out) once they exit the barrel which reduces muzzle energy as distance from the barrel increases and the shot pattern (BBs spread out) expands. Shotgun slugs are also very lethal, in fact may be excessive firepower for use in close-range situations. A study by Blackmore suggested that a projectile with 127 Joules (94 ft lbf (foot-pounds of force)) of kinetic energy (KE) is sufficient to penetrate the frontal bone of a three-year-old Angus cow [13]. Based upon the calculations of J of KE for the shotgun slug used in the study by Thomson, et al., the 12-gauge 2.75-inch one-ounce rifled slug had the equivalent of 5552 J (4095 ft lbf) of KE, which is more than 40 times the J of KE necessary to traverse the frontal bone of most steers [10]. Slugs are the preferred ammunition for circumstances that may require shooting from a distance.

A possible advantage of euthanasia using a shotgun is that within close range (three meters or 3.3 yd) and when properly directed, birdshot has sufficient energy to penetrate the skull, but is less likely to exit the skull. In the case of a free bullet from a high caliber firearm or a shotgun slug there is always the possibility of the bullet or slug exiting the skull creating a serious human health risk for the operator or by-standers. For safety reasons it is important that the muzzle of a shotgun (or any other firearm) never be held directly against the animal’s head. Discharge of the firearm results in the development of enormous pressure within the barrel that can result in explosion of the barrel and potential for injury of the operator and by-standers if the muzzle end is obstructed or blocked. It is important that anytime firearms are used, whether shooting from close range or a distance, the shooter is able to clearly visualize a safe backstop for bullets that miss or pass through their targets.

### 4.4. Captive Bolt

Captive bolt is a popular method of euthanasia for cattle in field situations; however, unlike euthanasia using a firearm, once the animal becomes unconscious, an adjunctive method to insure death must be applied. A study by Gilliam et al. found that approximately 10% of animals required a secondary adjunctive step to cause death [14]. In that study, euthanasia via a single shot with no adjunctive step successfully resulted in the death of 28/31 (90%), 17/19 (89.5%), 8/10 (80%) and 9/10 (90%) adult, young, neonate (penetrating captive bolt) and neonate (non-penetrating captive bolt) animals, respectively [14]. The clinical indicators of failure and need to apply an adjunctive method to cause death were a return to rhythmic breathing and prolonged time to cardiac arrest [14]. Specific reasons for the failure of penetrating captive bolt (PCB) to cause death include insufficient depth of penetration of the bolt, differences the resistance to bolt penetration related hardness and thickness of the skin and skull and the potential for slight misdirection of the shot. Gilliam et al. also found notable differences between shot placement locations and breed characteristics [14,15,16]. Therefore, an adjunctive method such as exsanguination, pithing or the intravenous injection of a saturated solution of potassium chloride (KCl) is highly recommended to ensure death whenever captive bolt is used [2,14,15,16].

Styles of PCB include an in-line (cylindrical) and pistol grip (resembling a handgun) design. Pneumatic (air-powered) captive bolt guns are primarily limited to use in slaughter plant environments, whereas gunpowder charged versions are used more often in on-farm environments. Depending upon model, the bolt may automatically retract or require manual re-placement back into the barrel through the muzzle.

Accurate placement of the captive bolt over the ideal anatomical site, energy (i.e., bolt velocity) and depth of penetration of the bolt determine effectiveness of the device to cause loss of consciousness and death. Bolt velocity is dependent upon the powder charge used and regular maintenance (i.e., cleaning) of the device. It is recommended that captive bolt guns be cleaned each time they are used with the same or similar solvents and lubricants used in the cleaning of firearms. Powder charges for captive bolt guns should be stored in airtight containers to prevent damage from the absorption of moisture that may occur in hot and humid conditions. Only penetrating captive bolt is recommended for euthanasia of adult cattle. Either the penetrating or non-penetrating captive bolt are appropriate for euthanasia of calves when followed by the use of an adjunctive (secondary step) method to assure death [2].

In contrast to the techniques described for gunshot, the animal must be restrained for accurate placement of the captive bolt. In addition, unlike use of a firearm, proper use of the captive bolt requires that the muzzle of the device be held firmly against the animal’s head over the intended site. Adjunctive methods should be implemented as soon as the animal is determined to be unconscious to avoid a possible return to consciousness. When conducting euthanasia by captive bolt, pre-planning and preparation are necessary to achieve the desired results [2].

## 5. Visual Indicators of Unconsciousness

The ability to distinguish an animal as conscious or unconscious is critical to assuring a humane death. Unconsciousness is broadly defined as a loss of individual awareness brought about by a disruption in the brain’s ability to integrate information. Anesthetic-induced unconsciousness in humans is defined as a loss in the ability to respond to verbal commands. In animals, the indication of unconsciousness associated with anesthesia is a loss of righting reflexes [17,18]. In cattle, unconsciousness occurring from brain damage by physical methods includes immediate collapse followed by tetanic spasms and involuntary hind limb movement [7,19,20]. There is an immediate loss of rhythmic breathing as well as palpebral and corneal reflexes. There should be no evidence of vocalization.

When it is necessary to euthanize a down cow, it is not possible to observe the immediate collapse of the animal associated with the instantaneous loss of consciousness. Therefore, it is necessary to monitor other clinical parameters. For safety’s sake, the person conducting the procedure should position himself/herself out of the range of legs and feet to check corneal reflexes and assure that the animal is unconsciousness. Once the animal is confirmed to be unconscious, the operator may choose to apply an adjunctive step or continue to monitor the animal for evidence of the possible return of a corneal reflex, righting reflex, respiration or vocalization. Monitoring should continue until death is confirmed.

## 6. Adjunctive Methods to Assure Death in Unconscious Animals

Potassium chloride, magnesium sulfate and magnesium chloride. While not acceptable as the sole method of euthanasia, the rapid intravenous administration of a saturated solution of potassium chloride (KCl), magnesium sulfate (MgSO_4_ or Epsom Salt) or magnesium chloride (MgCl_2_) may be used as adjunctive methods to assure death in unconscious animals. At 20 °C (68 °F) a saturated solution of KCl, MgSO_4_ or MgCl_2_ may be prepared by dissolving 342, 350 and 546 grams, respectively in one liter of water [21]. Potassium chloride is cardiotoxic and when administered by rapid intravenous injection causes cardiac arrest. Magnesium sulfate and MgCl_2_ are neuromuscular blocking agents and may not cause death as quickly as KCl. Normally, the administration of 120 to 250 mL of a saturated solution of KCl is sufficient to cause cardiac arrest and death in a mature bovine; however, one should be prepared to deliver more as needed. It is important to position oneself out of the reach of limbs and hooves that may cause injury during periods of involuntary movement. In most cases, it is safest to kneel down near the animal’s back and close to the animal’s head where one can reach over the neck to administer the injection into the jugular vein. Once the needle is in the vein, the injection should be delivered rapidly [2].

Exsanguination is usually accomplished via an incision of the ventral aspect of the throat or neck transecting skin, muscle, trachea, esophagus, carotid arteries, jugular veins and a multitude of sensory and motor nerves and other vessels. This procedure should never be used as a sole method of euthanasia. Exsanguination can be disturbing to observe due to the large volume of blood loss. When only the carotid arteries and jugular veins are cut, bleeding may persist at variable rates for 10–15 min or longer. Severing these vessels closer to the thoracic inlet where the vessels are larger will increase blood flow rate and reduce time until death [2].

Pithing is designed to cause death by increasing the destruction of brain, brainstem and the upper spinal cord tissue. It is performed by inserting a pithing rod through the entry site produced in the skull by the projectile (i.e., bullet or bolt). The operator manipulates the pithing tool to destroy brainstem and spinal cord tissue to ensure death. Muscular activity during the pithing process may be quite violent in some animals; therefore, persons conducting this procedure are advised to position themselves out of the reach of flailing legs and feet [2,22].

Second shot. Although one well-placed bullet or shot from a penetrating captive bolt is usually sufficient to cause immediate loss of consciousness with little likelihood of return to consciousness, one should always be prepared to deliver a second or even a third shot if necessary. When used as an adjunctive step, death is assured by the additional damage to brain tissue [2].

## 7. Anatomical Landmarks for Euthanasia of Cattle

The objective of euthanasia using the physical methods of gunshot and captive bolt are to cause sufficient damage to the brain to result in immediate loss of consciousness and death. Accomplishment of this objective requires the accurate delivery of a bullet or captive bolt at an anatomical site that is most likely to cause damage to the brainstem. At least two methods may be used to determine the proper anatomical site (See diagram in Figure 1) for conducting euthanasia with either a firearm or captive bolt. The method published in the 2013 AVMA Euthanasia Guidelines recommends that the point of entry for a projectile be at (or slightly above) the intersection of two imaginary lines, each drawn from the outside corner (lateral canthus) of the eye to the center of the base of the opposite horn (or where the horn would be in a horned animal [2]. A second method would be on the midline of the forehead half-way between two lines drawn laterally; one across the poll and the other between the outside corner (lateral canthus) of each eye [14,15,16,23].

## 8. Confirmation of Death

The confirmation of death should include assessment of the following parameters: lack of pulse, breathing, corneal reflex, failure of a response to firm toe pinch, graying of mucus membranes, and an inability to auscultate respiratory or heart sounds suggestive of cardiac arrest. Rigor mortis is one of the surest indicators of death [2]. Arguably, the most important clinical indicator of death is the lack of an auscultable heartbeat. Despite massive brain damage, it is common for the heart to continue to beat for seven to eight min or more in the immediate post-shot period. Heartbeat is controlled by the sino-atrial node (SA-node), a small body of specialized tissue in the right atrium of the heart that functions as the heart’s pacemaker. Within this tissue (SA-node) are cells with the ability to spontaneously produce an electrical impulse that travels through the heart via an electrical conduction system that causes contraction of the heart muscles. Disruption of the electrical potential within the heart muscle cells prevents their ability to contract, resulting in cardiac arrest.

## 9. Carcass Disposal

Proper disposal of carcass remains is important for many reasons. In North America, there are coyotes, buzzards and other scavenging animals willing to feed on carcass remains. In remote areas, this may seem a “natural way” to dispose of a deceased animal as it not only rids the premises of animal remains but also provides food for scavengers. However, this seemingly “natural way” may increase the potential for spread of infectious disease and in addition, it is unlawful and encourages scavengers to become predators when a carcass is not readily available. When animals have been euthanized with pentobarbital, exposed remains risk accidental poisoning of wildlife [24,25,26].

Part of the problem is that socially, economically and environmentally acceptable methods of carcass disposal have become increasingly difficult to find. There are fewer rendering companies than in the past and while some pay the owner for the pick-up and removal of livestock remains, today most charge for these services. Cost to renderers to process carcasses has increased in part because of the need to remove the brain, spinal cord and related tissues (i.e., specified risk material) due to concerns for Bovine Spongioform Encephalopathy (BSE). Since it is known that the causative agents of BSE (prions) concentrate in specified risk materials (SRM) most countries have imposed strict requirements on animal slaughter facilities and renderers to insure that these will not be used in human or animal food. This has caused some to look for other options for handling livestock mortalities.

Presently, choices for disposal of livestock carcasses includes rendering, burial, landfills, composting and incineration. In high livestock density areas (areas where there may be several large feedlots or dairies), animal mortalities and tissues discarded from the slaughter of animals are rendered into usable materials such as lard or tallow. Some of these products are utilized in the manufacturing of wet and dry pet foods. In February 2018, pentobarbital contaminated pet food was alleged to result in the death of a dog [27]. Subsequent testing by the Food and Drug Administration (FDA) (United States) confirmed the presence of pentobarbital in a tallow ingredient used in the dog food prompting an immediate nationwide recall of all potentially affected pet food products [28]. Although death or serious illness of animals consuming pentobarbital-contaminated pet food are unlikely, studies confirm that the pentobarbital will persist through the rendering process [29]. In a study designed to assess the distribution and fate of pentobarbital in rendered material from euthanized animals, researchers found it equally distributed between the meat and bone meal and tallow fractions [29].

Burial is a common method of carcass disposal and regulated by local laws and ordinances as to the number of pounds of animal carcass per acre per year that may be buried. It is best suited to geographic areas where the water table is deep and the soil is non-porous. Most regions have soil survey maps that can be used to determine the suitability of a specific site for carcass burial considering water table depth and soil porosity. Carcasses from euthanized animals can leach pentobarbital into the surrounding soil and ground water [30].

Landfills are a convenient and potentially cost effective way to dispose of animal mortalities where landfill operators will accept animal carcasses. Modern landfills are designed to protect the environment offering not only containment, but also a barrier to prevent wildlife access. Nonetheless, testing of water supplies near landfills has found detectable levels of pentobarbital up to 22 years following the time in which wastes were received [30].

Composting is a relatively inexpensive and environmentally sound approach to carcass disposal in conditions where compost piles can be properly managed. A recent study of horses euthanized with pentobarbital found that leachate from the decomposing carcass penetrated through the compost bed and into the soil beneath. Researchers concluded that it was not only important to provide a protective physical barrier to reduce the risk to scavenging by wildlife, but also recognize the risks of surface, groundwater and soil contamination [31].

Incineration would be expected to limit pentobarbital contamination concerns, but it is extremely expensive and mainly designed for smaller animals. Large animals may require dismemberment, which greatly adds to the inconvenience and cost of this method of disposal.

## 10. Penetrating Captive Bolt for Mass Depopulation

Currently, the only single step methods of euthanasia approved for use in cattle are gunshot and barbiturate overdose both of which are impractical for mass depopulation scenarios. Penetrating captive bolt is a third option; however, the need to apply an adjunctive step to assure death is impractical for depopulation of a large feedlot. For example, feedlots with 32,000 head or more of capacity market nearly 40% of cattle fed in the United States [32].

In the event of an epizootic disease outbreak involving a foreign animal disease in the United States, USDA APHIS personnel in conjunction with relevant federal, state, tribal, and local entities will be responsible to coordinate disease eradication plans in an attempt to prevent widespread losses. In preparation for such a possibility, USDA-APHIS proposed the development and validation of a portable pneumatically powered penetrating captive bolt equipped with a long bolt (i.e., 15 cm or six inches) and the option of low-pressure (15 psi or 103 kPa) air channel pithing through the bolt (AP-PCB). Two studies were conducted to validate the effectiveness of this prototype device under field conditions as a one-step method of euthanasia (e.g., not requiring an adjunctive step). The first study was conducted on 66 feedlot cattle using the AP-PCB [33]; the second study on 21 feedlot cattle; eight assigned to a group euthanized with the AP-PCB and 13 in a group euthanized with the Non-AP-PCB [34].

In both studies, cattle were comfortably restrained in a cattle-handling chute (crush) so that four leads of an electrocardiogram (ECG) could be applied and clinical variables safely and carefully monitored during the immediate post-shot period. The anatomic site used was that as described previously in this article under “Anatomic Landmarks for Euthanasia of Cattle”. A rope halter was applied to secure the head for accurate placement of the PCB. For cattle appearing to be apprehensive, a blindfold was used to reduce anxiousness. Immediately after the shot, researchers confirmed unconsciousness by assessing corneal and palpebral reflexes and by assuring, that rhythmic breathing had stopped. Following the confirmation of death, heads were disarticulated at the atlanto-occipital joint and taken to the Iowa State University College of Veterinary Medicine for further examination and scoring of brain trauma.

The results of both studies validated that the portable pneumatic penetrating captive bolt, whether fitted with the air-pithing or non-air pithing bolt, was effective for one-step euthanasia of feedlot cattle. Four animals in the first study required a second or third shot from the PCB because of technical errors (i.e., inadequate restraint that resulted in misdirection of the bolt) that resulted in the bolt missing the cranial vault. In the second study, the chute offered additional restraint options that improved handling of fractious animals and no animals required more than one shot from the PCB. When accurately placed over the correct anatomic site, the bolt entered the cranial vault causing an immediate loss of consciousness and death in all 87 (66 + 21 = 87) animals studied. Clinical observation in the immediate post-shot period revealed that none of the calves in either study exhibited corneal reflexes, righting reflexes, vocalization or respiration following the PCB. The mean time to death (i.e., cardiac arrest) was 7.3 min by auscultation and 8.3 min based upon electrical activity determined by ECG in study 1 and 7.3 min based upon ECG in the second study. Because of desire to avoid disruption of the ECG leads and because design of the chute limited access for auscultation, only ECG data were recorded in study 2 [33,34].

Postmortem examination of cattle in both studies revealed massive brain trauma precluding any possibility of a return to consciousness (Figure 2). The depth of penetration for a PCB used in packing plants is approximately 8.9 cm (3.5 inches). In these studies, mean bolt penetration depth was measured at 14.1 cm (5.5 inches) in study 1 and 15 cm (5.9 inches) in study 2. There were frequent fractures of the sphenoid and occipital bones, which make up the floor of the cranial vault. Direct damage to the thalamus/hypothalamus and brainstem was present in nearly all animals (Figure 2). There was no evidence of brain trauma attributable to the low-pressure pithing feature of the air-pithing bolt. It was concluded that the most important factors in successful use of this system were good restraint and accurate placement of the PCB over the proper anatomic site.

The above studies indicate that the depopulation of large numbers of animals is possible with existing technology. The more complicated issue with mass depopulation is carcass disposal. In some cases, it may be possible to depopulate large feedlots, dairies and beef cattle operations using commercial and private processing facilities. This assumes that affected cattle operations are reasonably close to packing plants and that movement does not present food safety concerns or risk further spread of disease. However, recognizing that depopulation would require good handling facilities and an infrastructure for processing and rendering of carcasses indicates that packers may be an important part of the overall depopulation plan. For smaller operations with fewer animals, mobile slaughter systems may be a consideration.

## 11. The Human Cost of Euthanasia

“Veterinarians are trained to help in nature’s healing process, but they also treat their patients with the knowledge that some of them will ultimately be slaughtered for human use. At other times, veterinarians are called upon to perform euthanasia, but not always for humane or medical reasons,” CS Manette [35].

Euthanasia is frequently the task of a veterinarian but certainly none of them pursued a career in veterinary medicine to secure an opportunity to euthanize animals. It is not the motivation that would drive someone to complete the arduous process of obtaining a veterinary degree. It is for most a love and desire to work with animals. The same is true for farmers and ranchers, shelter workers, laboratory researchers and animal technicians; virtually all whose lives or work involves daily contact with animals. However, intended or not, much of human interaction with animals causes them harm and oftentimes leads to their death.

## 12. The Caring and Killing Paradox

The requirement to perform euthanasia has been associated with multiple emotional and physical ailments including unresolved anger and grief, depression, sleeplessness, elevated blood pressure, and destructive behaviors such as substance abuse and suicide [36,37,38]. People experiencing or exhibiting these symptoms are said to be suffering from a form of work-related stress first labeled by Arluke as the “caring and killing paradox” [36,37]. It occurs among people whose life work is intended to promote the well-being of animals, but instead find themselves more often complicit in their death. One author characterizes this conflict between aspiration (the objective of promoting well-being) and reality (necessity to euthanize animals) as “moral stress” [1,38]. While intellectually people accept job positions as technicians in animal shelters, research laboratories and even veterinary clinics knowing that euthanasia is part of the job, it seems that many underestimate its psychological impact. Personnel directly engaged in the euthanasia of animals reported significantly higher levels of work stress and lower levels of job satisfaction [39].

The emotional issues stemming from caring and killing and moral stress are not unique to those who own and care for non-farm animals. They are also common to those who own and care for livestock and typified by statements such as the following:

“I have had to euthanize cows for producers who could not do it for themselves,” Many beef producers become very attached to the animals they care for, and they get a true satisfaction in caring for the animals in their possession. Dr. Jessica Laurin, DVM, Bovine Veterinarian/July–August 2007.

It is easy to delay euthanasia or avoid it altogether. A 2001 survey by the Canadian Food Inspection Agency of non-ambulatory cattle arriving at federally inspected slaughter plants and auction markets in Canada indicated that 89.8% were dairy cattle [40]. Survey data indicated that the majority of cattle developed the non-ambulatory condition on the farm of origin well-before being loaded for transport. Less than one percent became non-ambulatory as a consequence of injury during transit or upon arrival at their destination [40]. Whether animals with suspect mobility are truly transported by mistake or possibly, on purpose with the idea that some degree of compensation is better than none, fitness for travel should be carefully evaluated before animals are transported. Euthanasia may be the best option for animals with serious musculoskeletal disabilities and impaired mobility.

In the dairy industry, the market for male dairy calves varies with the demand for veal and/or beef feeder calves. In the past (during the very worst of times), with no market for male dairy calves the only economically viable option was to reduce losses by euthanizing male calves at birth. Fortunately, in recent time with the advent of sexed semen and the ability to sire more females than males, the market has been fair to good with fewer farms being forced to the extremes of euthanizing male calves. Jersey male calves are the notable exception. Because they are light and less heavily muscled, the veal and beef feeder market for Jersey male calves is poor. The use of beef sires on selected Jersey cows has improved the market for Jersey male calves by improving muscling and thus their marketability to the veal and feedlot cattle industries. The dairy goat industry has similar issues with the lack of a market for male kids. As a result most are euthanized at birth and not all by acceptable methods of euthanasia.

## 13. Compassion/Euthanasia Fatigue

Compassion fatigue (CF) is the depletion of internal emotional resources (i.e., emotional exhaustion) that results from caring for and helping traumatized or suffering people or animals [41]. By engaging in support of a person or animal that is suffering one opens the door to potential development of CF. Euthanasia is frequently connected with CF because it is considered a high risk factor. Compassion fatigue is huge issue for shelter workers and those who work in animal control. Although there are no specific published reports from farming systems, CF has multiple parallels with burnout, the “caring and killing” paradox, moral stress and Post-traumatic stress disorder (PTSD), all of which are common to the raising of livestock.

## 14. “Post-Traumatic Stress Disorder” (PTSD)

Post-traumatic stress disorder (PTSD) in humans [42] is a mental health condition triggered by experiencing or witnessing a terrifying event. Although commonly associated with soldiers who have experienced combat violence, PTSD may occur secondary to other life-threatening events such as an auto accident or sexual assault. Symptoms of PTSD often start within one month of the traumatic event or in some cases may not occur for several years after the incident. Disturbing memories of the experience that may occur as flashbacks or nightmares, feelings of negativity, hopelessness, guilt and overwhelming shame are just a few of a wide range of emotional problems one may experience [42].

A preliminary study at Purdue University College of Veterinary Medicine has found that the symptoms of PTSD are less intense among war veterans who have trained PTSD service dogs. The study consisted of 141 participants half of which were on the wait list to receive a service dog, the other half had a service dog. Results revealed that veterans suffering PTSD who had a service dog had better mental health. They experienced fewer symptoms of PTSD including less depression, higher degrees of life satisfaction, lower degrees of social interaction and less absenteeism from work compared to those who did not have a service dog [43].

In 2001, the Netherlands experienced an outbreak of foot and mouth disease that resulted in the culling of 270,000 animals [44]. A survey consisting of interviews was conducted 6–8 months after the crisis to assess the impact of this event on farmers and their families. Three regions were surveyed: Region (A) the areas where dairy farmer’s livestock had been culled; Region (B) areas where farmers were subject to severe restrictions, but no animals were culled, and Region (C) an area where there were no restrictions or culling. The study population consisted of 661 (51% of the dairymen contacted) Dutch dairy farmers: A = 215/370 (58%), B = 240/428 (56%) and C = 204/510. Results indicated that farmers that were older and had less education had the highest risk of PTSD symptoms. PTSD symptoms were highest in the culled area (Region A) and lowest in the free area (Region C). Nearly half of the farmers who had livestock culled and more than 20% of farmers in the restricted zone had PTSD and required professional help. Qualitative data indicated that the sight of their animals being slaughtered was engraved in their memory. Farmers described having flashbacks and nightmares as described for combat soldiers suffering from PTSD. While FMD only affected the animals, the consequences for farmers and their families were quite severe [45]. Similar observations were made in the United Kingdom [45]. Between 6.5 and 10 million animals were slaughtered across the UK. A 2003 report indicated that British farmers were more than twice as likely to consider suicide as people in other locations and the previous FMD crisis was believed to be at least partly to blame [46].

## 15. Perpetuation-Induced Traumatic Stress (PITS)

Perpetration-induced traumatic stress (PITS) is a variant on PSTD reported in the scientific literature in 2002 [47]. Prior to that time, the study of PTSD was focused primarily on soldiers of the Vietnam War era and others who were the victims of trauma. Subsequent research on combat veterans found that the severity of PTSD was greater if they actually killed humans or believed their actions resulted in human death or if they were directly involved in combat-related atrocities [47]. Similar findings were observed of soldiers returning from the Operation Iraqi Freedom [48]. Perpetration-induced traumatic stress (PITS) has also been identified in individuals who participate in the killing of healthy animals and has many of the similarities of moral stress [49]. Researchers surveyed 148 veterinarians, veterinary nurses, animal shelter staff, and animal research staff and found that 11% of the animal care workers were experiencing moderate levels of traumatic stress due to their experiences with euthanasia of animals. More than half of survey participants found euthanasia their least desired task [49].

Circumstances that may require the necessity to euthanize large numbers of animals (as in a depopulation scenario) can be extremely difficult even for those with work experience in abattoirs and managing welfare disasters. Canadian researchers reported on an economic downturn in the pig market that brought about the need to kill thousands of healthy surplus piglets in Manitoba, Canada [50]. Despite efforts of the participants to rationalize and distance themselves from the moral responsibilities of their actions, several developed mental health problems with one ultimately requiring hospitalization [50]. Similar to the PITS of combat soldiers, the life of the veterinarian is never directly threatened by the killing of animals; only their identity as a veterinarian.

Mental illness characterized by depression and anxiety is prevalent for those required to participate in the euthanasia of animals irrespective of their background or level of preparation. Euthanasia is difficult and the greatest cost to humans may not be measurable in dollars and cents.

## 16. In Summary

The methods of euthanasia considered to be unacceptable are those known to cause pain and distress. Some resort to these for reasons of convenience or because they find euthanasia by firearm or captive bolt aesthetically/emotionally disturbing. The acceptable methods of euthanasia for cattle include overdose of an anesthetic, gunshot and captive bolt with appropriate adjunctive methods to assure death. The most commonly used injectable anesthetic is pentobarbital. It is preferred in sensitive conditions because of its ability to provide a smooth transition from consciousness to unconsciousness and death. However, there are multiple deterrents to the use of pentobarbital for euthanasia of cattle. A major complication is that of carcass disposal. Failure to properly dispose of the remains of deceased animals euthanized with pentobarbital risks potential poisoning of wildlife. Pentobarbital also persists through rendering and composting leaving few options for the disposal of carcass remains. The use of a firearm or captive bolt avoids some of the complications encountered with pentobarbital. Selection of the appropriate caliber of firearm and bullet or shotshell are important when gunshot is used for euthanasia. Captive bolt also requires selection of the appropriate powder charge and choice of a penetrating or non-penetrating (concussion style mushroom head) bolt. One of the most important factors in successful euthanasia is use of an anatomic site that is most likely to cause an immediate loss of consciousness and death. In cattle, the anatomic site or point of entry for a projectile should be at (or slightly above) the intersection of two imaginary lines, each drawn from the outside corner (lateral canthus) of the eye to the center of the base of the opposite horn (in a polled animal where the horn would normally be if the animal had horns). A second method would be on the midline of the forehead half-way between two lines drawn laterally: one across the poll and the other between the outside corner of each eye (lateral canthus). When captive bolt is used, a knowledge of the indicators of unconsciousness is critical since adjunctive methods to assure death must never be applied in conscious animals. Adjunctive or secondary methods that may be applied to assure death in unconscious animals includes the intravenous administration of a saturated solution of KCl or MgSO_4_, exsanguination, pithing or a second shot. Scenarios that may require mass depopulation of a large feedlot, dairy or cattle operation require a system that assures death without the need for an adjunctive step. A pneumatic penetrating captive bolt has been developed and validated as a one-step euthanasia device for use in animal health emergencies. Finally, the need to euthanize animals whether it be on an individual basis or in a mass depopulation scenario is stressful and can lead to significant emotional and psychological difficulties for people involved.

## 17. Conclusions

Euthanasia is sometimes the only solution for relief of otherwise uncontrollable pain and distress. In cattle, there are three methods: the overdose of an anesthetic, gunshot or a captive bolt with adjunctive steps to assure death. Understanding how to apply these techniques reduces the possibility of pain and distress in animals that may require euthanasia. It is also a prerequisite to dealing with the emotional stress that accompanies the task of euthanasia.

## Figures and Tables

**Figure 1 animals-08-00057-f001:**
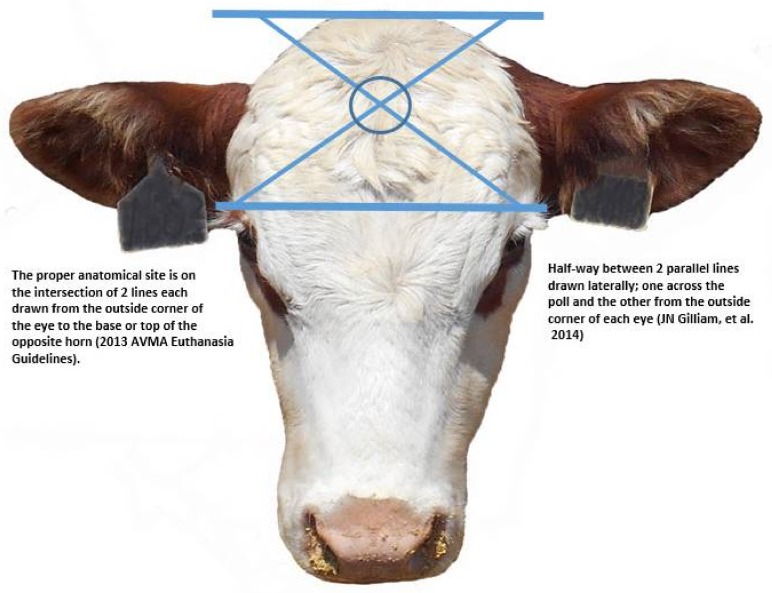
Anatomic sites for conducting method using the physical methods of gunshot and captive bolt. The above photo and captions identify two methods for determining the proper anatomical site for conducting euthanasia procedures in cattle [2,14].

**Figure 2 animals-08-00057-f002:**
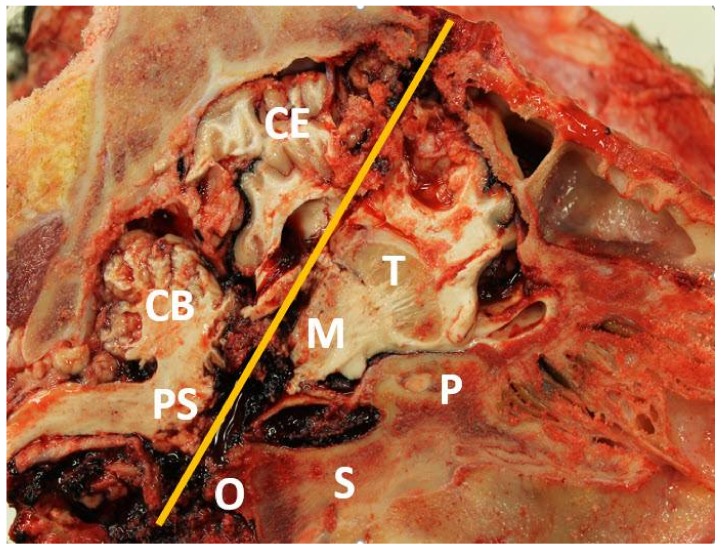
Sagittal section of the skull of a bovine euthanized with a prototype pneumatic penetrating captive bolt designed for use in mass depopulation of a large feedlot, dairy or beef cattle operation. Labelled structures are as follows: Cerebrum (CE), Cerebellum (CB), Thalamus (T), Midbrain (M), Pons (PS), Presphenoid bone (P), Sphenoid bone (S) and Occipital bone (O).

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
