# Peer review of "Euthanasia of Cattle: Practical Considerations and Application"

_animals, 2018, doi:10.3390/ani8040057_

Round 1

Reviewer 1 Report

Line No.

Clause

Concern

Weighting*

77

Part of the problem is that socially…

BSE should be mentioned, major changes in meat and bone meal use

2

126

Calibers ranging from .32 to .45 are recommended

The .22 rimfire is a pistol cartridge although widely used in rifles

1

129-131

The .22 caliber handgun is not recommended for routine euthanasia of   cattle regardless of the type of bullet used, because of the inability to   consistently achieve desirable muzzle energies with standard commercial   loads.

A statement this strong should have a reference.  I agree with the factuality with bullet   performance variance and barrel length differences the advertised bullet energy   on the box is for rifles with at least 26 inches of barrel to accelerate in.   A 3 inch barrel Saturday night special is completely undependable(1).

Reference 13, 14 may be suitable

2

151

disperse as it leaves the end of the gun barrel

The shot exits as a single metal bolus and then rapidly spreads out   losing energy as the face of the bolus expands.  Disperses is a inaccurate word to describe this   process.

2

156

in close-range situations

Over penetration is a serious human health risk

3

157

shooting from a distance.

shooting from a distance with a clearly visible safe backstop

2

*1=may be reviewer prejudice and does not substantially reflect the quality of the paper (preference)

2 = Reviewer would like to see this change if editor and authors agree

3 =  Reviewer would like to see this change

Non Specific Comments:

The novel information in this paper is the concept of “pneumatically pithing cattle”. An indication of whether this technology is currently commercialized would be of great interest to this reader. An image of the cross-section of a pneumatically pithed animal would also be of great value. 

Figure 1 is excellent.

References

1.            Suddaby L, Weir B, Forsyth C. The Management of .22 Caliber Gunshot Wounds of the Brain: A Review of 49 Cases. Canadian Journal of Neurological Sciences / Journal Canadien des Sciences Neurologiques. 1987;14(3):268-72.

Author Response

Reviewer 1

Line 77                  Part of the problem is that socially… BSE should be mentioned, major changes in meat and bone meal use         2

The following has been added:

Cost to renderers has increased in part because of the need to remove the brain, spinal cord and related tissues (i.e. specified risk material) due to concerns related to Bovine Spongioform Encephalopathy (BSE).  Since it is known that the causative agents of BSE (prions) concentrate in specified risk materials (SRM) most countries have imposed strict requirements on animal slaughter facilities and renderers to insure that these will not be used in human or animal food. 

Line 126                Calibers ranging from .32 to .45 are recommended, The .22 rimfire is a pistol cartridge although widely used in rifles        1

For the purposes of euthanasia, accuracy is improved when handguns are limited to close-range shooting within 1 meter (3.3 feet) or less of the intended target.

129-131                The .22 caliber handgun is not recommended for routine euthanasia of cattle regardless of the type of bullet used, because of the inability to   consistently achieve desirable muzzle energies with standard commercial loads. 

A statement this strong should have a reference.  I agree with the factuality with bullet   performance variance and barrel length differences the advertised bullet energy   on the box is for rifles with at least 26 inches of barrel to accelerate in.   A 3 inch barrel Saturday night special is completely undependable(1).

Suggested revision:

The .22 caliber handgun is not recommended for routine euthanasia of adult cattle regardless of the type of bullet used, because of the inability to consistently achieve desirable muzzle energies with standard commercial loads.  Muzzle velocity and energy of a bullet are generally higher when fired from a rifle.  A longer barrel allows the propellant (gunpowder) to burn more completely which maximizes velocity of the bullet as it leaves the muzzle.  The shorter barrel of a handgun results in reduced muzzle velocity since much of the pressure propelling the bullet through the barrel is dissipated into the air prior to complete burning of the gunpowder.  A study by Thomson et al. found that the 9 mm pistol firing a metal-jacketed bullet performed poorly in a study involving bovine cadaver skulls.  Despite an estimated KE of 427.85 J with this firearm and bullet combination, it caused the least amount of brain tissue and brainstem trauma.  One might speculate that a shorter barrel length may be part of the explanation.  At least one source indicates that heavier rounds with larger propellant loads such as those used in 9 mm and .45 caliber pistols require a barrel length of 16 inches or more for bullets to reach maximum muzzle velocity (Bullet Velocity and Barrel Length).  Ultimately, the damage potential of a firearm and bullet/shotshell combination is determined by multiple factors including thickness of the skin and skull as well as characteristics of the projectile such as a hollow point or a full metal jacket (Thomson et al ).     

Reference 13, 14 may be suitable                                                                                              2

151                         Birdshot begins to disperse as it leaves the end of the gun barrel; however, if the operator stays within short range of the intended anatomic site, the birdshot will strike the skull as a compact bolus or mass of BBs with ballistic characteristics on impact and entry that are similar to a solid lead bullet.  At close range, penetration of the skull is assured with massive destruction of brain tissue from the dispersion of birdshot into the brain that results in immediate loss of consciousness and rapid death. 

Shotguns loaded with buckshot or birdshot (#4 shot) are appropriate for euthanasia of cattle when used from a distance of 3 m (3.3 yd). Although all shotguns are lethal at close range, the preferred gauges for euthanasia of mature cattle are 20, 16, or 12. The BBs in birdshot exit the barrel as compact bolus or mass with ballistic characteristics on impact with  the skull at close range that are similar to those of a solid lead bullet.  Penetration of the skull is assured with massive destruction of brain tissue from the dispersion of birdshot into the brain that results in immediate loss of consciousness and rapid death.  A shotgun loaded with birdshot is less effective when shooting from a distance because the BBs in a shotshell will begin to scatter once they exit the barrel (i.e. spread out) which reduces muzzle energy as distance from the barrel increases and the shot pattern (BBs spread out) expands.  Shotgun slugs are also very lethal, in fact considered to be excessive firepower for use in close-range situations.  Slugs are the preferred ammunition for circumstances that may require shooting from a distance.   2

156                         in close-range situations - Over penetration is a serious human health risk              3

In the case of a free bullet from a high caliber firearm or a shotgun slug there is always the possibility of the bullet or slug exiting the skull creating a serious human health risk for the operator or by-standers. 

157                         shooting from a distance.  shooting from a distance with a clearly visible safe backstop                                    2

It is important that anytime firearms are used, whether shooting from close range or a distance, the shooter is able to clearly visualize a safe backstop for bullets that miss or pass through their targets. 

Non Specific Comments:

The novel information in this paper is the concept of “pneumatically pithing cattle”. An indication of whether this technology is currently commercialized would be of great interest to this reader. An image of the cross-section of a pneumatically pithed animal would also be of great value.

A sagittal section has been added as Figure 2 – the Figure is at the end of the document – editor is free to reposition as desired.

Figure 1 is excellent.

Thank you.

References

1.       Suddaby L, Weir B, Forsyth C. The Management of .22 Caliber Gunshot Wounds of the Brain: A Review of 49 Cases. Canadian Journal of Neurological Sciences / Journal Canadien des Sciences Neurologiques. 1987;14(3):268-72.

Thank you for the suggestion, I reviewed this paper, but did not use this reference. 

Reviewer 2 Report

This was a well written review on euthanasia of cattle. It would be improved by making it clear in the abstract what the purpose of this review is.  Also, it would be helpful to make it clearer what the main topics within the manuscript are. Suggest that the manuscript could be separated into 6 main headings with the rest as sub-headings (Introduction, Unacceptable methods of euthanasia, Acceptable methods of euthanasia, Mass depopulation and Summary)

L15 – Is it possible to add an age range for calves/young animals?

L19-Suggest putting beef before cattle

L23-Recommend including the purpose/aim of the manuscript at the beginning of the abstract.

L51-Chapter?

L60-61-Recommend moving Unacceptable methods section here (L251-273) and giving it the title ‘Unacceptable methods of euthanasia’

L61-Recommend including a main heading ‘Acceptable methods of euthanasia’

L80-Add ‘for’ before other

L125-Check the use of abbreviations for units throughout.

L127-Delete ‘a’ before solid

L139-What does LR mean?

L143-.22 LR

L155-165-Add a reference for this statement

L170-Why did 10% animals need a secondary step – fault in the captive bolt or the use of it?

L260-Inhumane

L262-Change is to maybe

L269-Suggest changing to ‘..;none of the animals achieved an aesthetic state based upon EEG. Also, spell out EEG.L296-300-Please include age range, breed and gender of the animals used in these 2 studies.

L303-Add beef before cattle

Author Response

Reviewer 2

This was a well written review on euthanasia of cattle. It would be improved by making it clear in the abstract what the purpose of this review is.  Also, it would be helpful to make it clearer what the main topics within the manuscript are. Suggest that the manuscript could be separated into 6 main headings with the rest as sub-headings (Introduction, Unacceptable methods of euthanasia, Acceptable methods of euthanasia, Mass depopulation and Summary)

I appreciate the reviewer’s comments – the abstract is limited to 200 words.  My objective was to touch on all issues to be discussed in the chapter. Although brief, I have tried to provide a clearer indication and justification for this article in the Introduction.  I have also expanded the section on the impact of euthanasia on human emotional/mental health.

L15 – Is it possible to add an age range for calves/young animals?

3 months of age or less

L19-Suggest putting beef before cattle

done

L23-Recommend including the purpose/aim of the manuscript at the beginning of the abstract.

I have expanded the introduction to include a better discussion of the purpose/aim

L51-Chapter?

Yes, this is mistake – this author was requested to offer this contribution and was under the mistaken impression that it was to be chapter rather than a review paper on euthanasia of cattle.

L60-61-Recommend moving Unacceptable methods section here (L251-273) and giving it the title ‘Unacceptable methods of euthanasia’

The paper has been reorganized and the Unacceptable methods precedes the Acceptable methods

L61-Recommend including a main heading ‘Acceptable methods of euthanasia’

This change has been made

L80-Add ‘for’ before other

Done

L125-Check the use of abbreviations for units throughout.

Done

L127-Delete ‘a’ before solid

Done

L139-What does LR mean?

LR means Long Rifle – this has been added in parentheses

L143-.22 LR

.22 LR means .22 Long Rifle

L155-165-Add a reference for this statement

It is suggested that a projectile with 127 Joules (94 ft lbf {foot-pounds of force}) of kinetic energy (KE) is sufficient to penetrate the frontal bone of a 3-year-old Angus cow (Blackmore, 1985).  Based upon the calculations of J of KE for the shotgun slug used in the study by Thomson,et al., the 12-gauge 2.75-inch 1-ounce rifled slug had the equivalent of 5,552 J (4095 ft lbf) of KE, which is more than 40 times the J of KE necessary to traverse the frontal bone of most steers.

Blackmore DK. Energy requirements for the penetration of heads of domestic stock and the development of a multiple projectile. Vet Rec 1985;116:36–40.

Thomson DU, Wileman BW, Rezac DJ, Miesner MD, Johnson-Neitman JL, Biller DS.  Computed tomographic evaluation to determine efficacy of euthanasia of yearling feedlot cattle by use of various firearm-ammunition combinations.  Am J Vet Res 2013;74:1385–1391.

L170-Why did 10% animals need a secondary step – fault in the captive bolt or the use of it?

Captive bolt is a popular method of euthanasia for cattle in field situations; however, unlike euthanasia using a firearm, once the animal becomes unconscious, an adjunctive method to insure death must be applied.  A study by Gilliam et al. found that approximately 10% of animals required a secondary adjunctive step to cause death [15]. In that study, euthanasia via a single shot with no adjunctive step successfully resulted in the death of 28/31 (90%), 17/19 (89.5%), 8/10 (80%), and 9/10 (90%) adult, young, neonate (penetrating captive bolt) and neonate (non-penetrating captive bolt) animals, respectively (Gilliam et al., 2014) .  The clinical indicators of failure and need to apply an adjunctive method to cause death were a return to rhythmic breathing and prolonged time to cardiac arrest (Gilliam et al, 2014).  Specific reasons for the failure of PCB to cause death include insufficient depth of penetration of the bolt, differences the resistance to bolt penetration related hardness and thickness of the skin and skull and the potential for slight misdirection of the shot.  Gilliam et al., also found notable differences between shot placement locations and breed characteristics (Gilliam et al, 2018).  Therefore, an adjunctive method such as exsanguination, pithing or the intravenous injection of a saturated solution of potassium chloride (KCl) is highly recommended to ensure death whenever captive bolt is used [1,15,16].  

Gilliam JN; Woods J; Hill J, Shearer JK, Reynolds J, Taylor JD.  Evaluation of the Cash Euthanizer Captive Bolt System as a Single Step Euthanasia Method for Cattle of Various Ages.  Poster presented at the 4th International Symposium on Beef Cattle Welfare, Iowa State University, Ames, IA, July 16-18, 2014

L260-Inhumane

Done

L262-Change is to maybe

Done

L269-Suggest changing to ‘..;none of the animals achieved an aesthetic state based upon EEG. Also, spell out EEG.L296-300-Please include age range, breed and gender of the animals used in these 2 studies.

These changes have been made and all available information on the animals used in the ISU study incorporated into this as requested.

L303-Add beef before cattle

This section has been revised.

Reviewer 3 Report

Delete lines 45-49

Rewrite lines 49-50 to be the beginning of the introduction. Consider citing welfare literature to strengthen your argument

Lines 53-55: streamline this sentence

Lines 56-57: I assume you are referring to the person performing the euthanasia-specify this and why it is important (cite compassion fatigue lit)

Lines 57-60: delete or rephrase

Line 68: change title to just “Carcass Disposal”

Line 70: delete “plenty of”

Line 83: can you define “high” livestock density?

Line 93: add “is” before “regulated”

Line 97: are there regulations or laws prohibiting burial of carcasses from euthanized animals?

Line 99: do most landfills allow carcasses? Do they have restrictions for euthanized animals?

Line 114: comma after “Gunshot” and lowercase “A”

Line 157: define the minimum distance recommended

Line 161: change to “risk of injury”

Line 180: how frequently, typically, do captive bolts need to be cleaned?

Line 196: are you referring to the AVMA Euth Guidelines? Please specify and cite

Line 213: change to “causes”

Line 215: are you referring to adult cows? Line

Line 244: specify and cite the guidelines you are referring to

Lines 252-258: cite where this information is from

Lines 259-261: repetitive, delete this sentence

Delete lines 262-264

Line 272-273: why is it considered unacceptable? Cite source

Lines 279-282: condense these two sentences into one, and also explain why the SA node continues to fire

Line 290: change to “an adjunctive step” (instead of “and”)

Line 295: please state year of study

Lines 315-334: delete those lines and start this section with the “caring and killing paradox.”

Lines 337-339: needs citation

Lines 340-348: delete those lines. Focus on discussing research about euthanasia and mental health in this section. You can discuss compassion fatigue in small animal practice if needed, and speculate on the link to farm animal euthanasia, but most of the commentary borders on pedantic and is not appropriate for this article

Lines 352-382: delete this section. It is very repetitive and does not add to the information you have presented, and is too long for a conclusion.

This is basically just a summary of the AVMA Euthanasia Guidelines. More, novel information and ideas for creating and implementing mass depopulations would make this a much more useful paper.

Author Response

Reviewer 3

Delete lines 45-49

Done

Rewrite lines 49-50 to be the beginning of the introduction. Consider citing welfare literature to strengthen your argument

This section has been completely revised

Lines 53-55: streamline this sentence

This section has been completely revised

Lines 56-57: I assume you are referring to the person performing the euthanasia-specify this and why it is important (cite compassion fatigue lit)

This section has been completely revised

Lines 57-60: delete or rephrase

This section has been completely revised

Line 68: change title to just “Carcass Disposal”

This has been done and this section has been moved to a occur later in the paper

Line 70: delete “plenty of”

Done

Line 83: can you define “high” livestock density?

Areas where there are numerous animals on large feedlots or dairies  -

Line 93: add “is” before “regulated”

I believe this reads best as it is written

Line 97: are there regulations or laws prohibiting burial of carcasses from euthanized animals?

Yes, as described there are regulations in some areas where mass burial or burial of large numbers of animals might risk ground water contamination. 

Line 99: do most landfills allow carcasses? Do they have restrictions for euthanized animals?

Some do and some don’t – for those that do accept animal carcasses most will have limits or restrictions on what they will accept.

Line 114: comma after “Gunshot” and lowercase “A”

This section has been completely revised

Line 157: define the minimum distance recommended

3 m or 3.3 yd.

Line 161: change to “risk of injury”

This section has been completely revised

Line 180: how frequently, typically, do captive bolts need to be cleaned?

After each use - This has been added

Line 196: are you referring to the AVMA Euth Guidelines? Please specify and cite

The 2013 AVMA Guidelines have been cited

Line 213: change to “causes”

This has been changed

Line 215: are you referring to adult cows? Line

Yes, this has been added

Line 244: specify and cite the guidelines you are referring to

Added the acronym AVMA

Lines 252-258: cite where this information is from

It is cited

Lines 259-261: repetitive, delete this sentence

I believe this reads differently now

Delete lines 262-264

I quite strongly disagree.  As Chair of the Food Animal Working Group of the AVMA’s Panel on Euthanasia, I know this to be a major problem with euthanasia of cattle and other animals at least in the United States.  I was a part of the Dewell et al. study that established that it was not possible to reach an anesthetic state with xylazine.  If the editor would like to alter the wording I am OK with it as long as the point of this is not lost.   

Line 272-273: why is it considered unacceptable? Cite source

Since an animal administered Xylazine hydrochloride (XH) does not reach an unconscious (anesthetized) state and the use of potassium chloride (KCl) causes cardiac arrest, this does not meet the criteria to be considered euthanasia.  Cardiac arrest in a tranquilized animal is considered to be a painful, inhumane death.   There are 3 references cited in the manuscript:  Dewell et al, 2013; Goodman and Gilman, 2006; and the 2013 AVMA Euthanasia Guidelines.

Lines 279-282: condense these two sentences into one, and also explain why the SA node continues to fire

I have added an additional explanation on the SA-node 

Line 290: change to “an adjunctive step” (instead of “and”)

Done

Line 295: please state year of study

This section has been completely revised

Lines 315-334: delete those lines and start this section with the “caring and killing paradox.”

This section has been completely revised

Lines 337-339: needs citation

This section has been completely revised

Lines 340-348: delete those lines. Focus on discussing research about euthanasia and mental health in this section. You can discuss compassion fatigue in small animal practice if needed, and speculate on the link to farm animal euthanasia, but most of the commentary borders on pedantic and is not appropriate for this article

This section has been completely revised

Lines 352-382: delete this section. It is very repetitive and does not add to the information you have presented, and is too long for a conclusion.

This section is a summary not a conclusion.  The conclusion is only 5 lines in length (lines 391-396) – the editor may delete the summary if desired.

This is basically just a summary of the AVMA Euthanasia Guidelines. More, novel information and ideas for creating and implementing mass depopulations would make this a much more useful paper. 

The section on mass depopulation has been revised to include additional detail on our research at ISU.  I have also expanded the section on impact on human emotional states.